# RuDar: Weather Radar Dataset for Precipitation Nowcasting with Geographical and Seasonal Variability

## Abstract

Precipitation nowcasting, a short-term (up to six hours) rain prediction, is arguably one of the most demanding weather forecasting tasks. To achieve accurate predictions, a forecasting model should consider miscellaneous meteorological and geographical data sources. Currently available datasets provide information only about precipitation intensity, vertically integrated liquid (VIL), or maximum reflectivity on the vertical section. Such single-level or aggregated data lacks description of the reflectivity change in vertical dimension, simplifying or distorting the corresponding models.

To fill this gap, we introduce an additional dimension of the precipitation measurements in the RuDar dataset that incorporates 3D radar echo observations. Measurements are collected from 30 weather radars located mostly in the European part of Russia, covering multiple climate zones. Radar product updates every 10 minutes with a 2 km spatial resolution. The measurements include precipitation intensity (mm/h) at an altitude of 600 m, reflectivity (dBZ) and radial velocity (m/s) at 10 altitude levels from 1 km to 10 km with 1 km step. We also add the orography information as it affects the intensity and distribution of precipitation. The dataset includes over 50 000 timestamps over a two-year period from 2019 to 2021, totalling in roughly 100 GB of data.

We evaluate several baselines, including optical flow and neural network models, for precipitation nowcasting on the proposed data. We also evaluate the uncertainty quantification for the ensemble scenario and show that the corresponding estimates do correlate with the ensemble errors on different sections of data. We believe that RuDar dataset will become a reliable benchmark for precipitation nowcasting models and also will be used in other machine learning tasks, e.g., in data shift studying, anomaly detection, or uncertainty estimation. Both dataset and code for data processing and model preparation are publicly available [1].

## 1 Introduction

Precipitation nowcasting is the task of forecasting a rainfall situation (precipitation location and strength) for a short period of time, usually up to six hours. Due to climate change the frequency and magnitude of extreme weather events, e.g. sudden downpours, increase, and the techniques for forecasting such events are needed. Precipitation nowcasting can provide information about such events with a high spatiotemporal resolution. Such kind of weather forecasting plays an essential role in resource planning in the agricultural industry, aviation, sailing, etc. as well as in daily life.

Incorrect precipitation forecasting could have a negative impact on human life activity, and data with diverse meteorological and geographical characteristics are needed for improving precipitation nowcasting models. The different benchmark dataset usage could improve the quality of precipitation nowcasting models to minimize the risk of forecasting error.

Previously published benchmarks Holleman (2007); Shi et al. (2017); Ansari et al. (2018); Ramsauer et al. (2018); Franch et al. (2020); Veillette et al. (2020) provide data collected with one or several

---

[1]URL is hidden for the blind review.

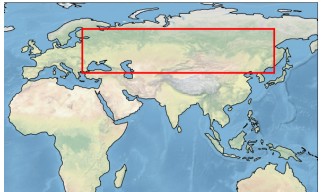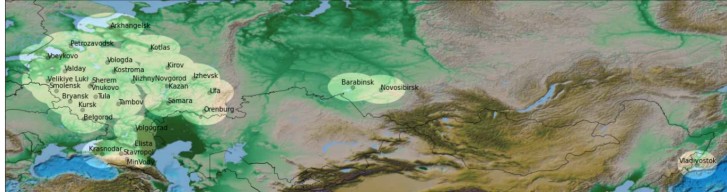

Figure 1: The geographical area covered by the proposed weather radar dataset (light areas). The covered area has a variety of geographic and climatic characteristics. The color indicates the height above sea level.

weather radars. Some of those datasets only contain information about precipitation intensity, others provide vertically integrated liquid value (VIL) or maximum reflectivity on the vertical section.

However, a single measurement type is often not enough for extreme weather events forecasting. Thereby, we propose a RuDar dataset that contains several measurement products: reflectivity (dBZ) and radial velocity (m/s) on ten altitude levels from 1 km to 10 km with 1 km step and intensity (mm/h) on a 600 m altitude level. Each measurement was carried out with a 2 km spatial resolution and a 10 minute temporal resolution. The dataset from 30 dual-pol Doppler weather radars were collected and processed at the Radar Center of the Central Aerological Observatory (CAO) of the Russian Federal Service for Hydrometeorology and Environmental Monitoring (ROSGIDROMET) and is used by our team within the conditions of commercial contract. For each radar, we additionally provide information about the surrounding orography Becker et al. (2009). The radars are located mostly in the European part of Russia as shown in Figure 1, therefore, a wide range of geographical and climatic conditions is considered. The proposed dataset includes more than 50 000 timestamps over a two year period from 2019 to 2021, allowing to investigate the effect of seasonality on rainfall forecast.

We illustrate the applicability of our dataset to the nowcasting task by benchmarking the current state-of-the-art optical flow approach Ayzel et al. (2019) and neural network models Shi et al. (2015); Veillette et al. (2020); Ravuri et al. (2021) on it. Experiments show a seasonal dependence having effect on the algorithm performance due to different precipitation intensity rates and differences between adjacent timestamps in different months.

The main paper contributions are (i) published weather radar dataset with different geographical and climatic conditions (provided under the CC BY NC SA 4.0 license) together with the accompanying exploratory data analysis, (ii) evaluations of common simple precipitation nowcasting models and its extenstion to support additional data, (iii) uncertainty estimation and its connection to the error for the nowcasting ensemble case, and (iv) accompanying source code for data processing and experiments.

The structure of the paper is as follows: Section 2 covers previously published datasets for the precipitation nowcasting task, Section 3 describes the proposed dataset, Section 4 introduces evaluated nowcasting benchmarks, Section 5 explores the uncertainty estimation scenario for the ensemble of models, and Section 6 concludes the paper.

## 2  RELATED WORK

Doppler weather radar is the most effective tool for detecting precipitation. The radar measures reflectivity of radio waves from precipitation drops, which can then be converted into precipitation intensity using Z-R relation  Marshall & Palmer (1948). Standard ways of obtaining a single reflectivity measure from the different heights is either taking measurements from only the lower level (base reflectivity), or aggregating these measurements by the maximum value (composite reflectivity). In addition, a Doppler radar can detect movement towards or away from itself, which allows measuring the speed of precipitation movement along or against the direction of the radar. The latter type of measurement is called radial velocity.

In the public domain, one can find quite a variety of weather radar datasets collected and maintained by international agencies. We summarized the information about some of them in Table 1, where we

provided a comparison by spatial, temporal and pixel resolution, time periods, geographic coverage and number of radars from which measurements were taken.

The first three data sources in Table 1 are more large databases than ready-made ML benchmarks:

- **NEXRAD** Ansari et al. (2018) by US National Oceanic and Atmospheric Service (NOAA), which has been collected since 1994 and contains reflectivity data, radial velocity, and derivative products.
- **MRMS** Zhang et al. (2016) by NOAA National Severe Storms Laboratory combines the previous source together with various sources to provide severe weather, transportation, and precipitation products with 2-minute updates. This dataset is used in a number of nowcasting studies Sønderby et al. (2020), Klocek et al. (2021).
- **Radarnet** Fairman et al. (2017) by UK Met Office, which has been collected since 1970 and contains composite precipitation data derived from reflectivity data.

The ready-to-go datasets, suitable for exploration by ML practitioners and researchers include:

- **HKO-7** Shi et al. (2017) by Hong Kong Observatory (HKO) contains a single-height reflectivity level from a single radar at the center of Hong Kong for a six years period.
- **KNMI** Overeem & Imhoff (2020) by the Royal Netherlands Meteorological Institute (KNMI) contains ten-year single-height composite reflectivity data from two radars.
- **TAASRAD19** Franch et al. (2020) by Meteotrentino contains nine-year aggregated reflectivity data from a single radar located in the Italian Alps. It is interesting due to geographical specifics and a large amount of extreme phenomena, such as snowstorms, hails, downpours, etc.
- **SEVIR** Veillette et al. (2020) by NOAA and the Geostationary Environmental Satellite System (GOES) contains US radar and satellite data for a two year-period, sampled either randomly or on an event basis. It is focused on the detection of storm events.
- **RADOLAN** Ramsauer et al. (2018) by the German Weather Service contains three-year reflectivity and precipitation data collected by 18 radars in Germany.

Our dataset was collected by 30 radars of the Central Aerological Observatory (CAO) and contains two years of observations (spring 2019 – spring 2021). Measurements were carried out mainly in the European part of Russia, but some areas of the Siberian and Far Eastern regions were also captured.

The main advantages of our dataset relative to the abovementioned are as follows:

1. A large area is covered with different climatic conditions – from the extreme north (Arkhangelsk, $64.62° \, N \, 40.51° \, E$ ) to the southern regions (Krasnodar, $45.04° \, N \, 39.15° \, E$ ).

2. Dataset introduces continuous measurements over a year period, which allows taking into account the influence of seasonal dependence on strength and distribution of precipitation.

3. The data presents several radar products at once: reflectivity and radial velocity at 10 altitude levels (1 – 10 km) and precipitation (600 m). We do not specifically aggregate the reflectivity and radial velocity data, as we believe that it can be useful for forecasting to know how moisture and its velocity are distributed depending on the height. There are situations when precipitation has not yet appeared at low levels, but at the same time, at high levels, there is information about future precipitation (see an example in Figure 6 in Supp.Mat.).

4. In addition to the radar data itself, we provide information about the orography in the area around each radar. These data allow one to investigate the influence of the surrounding orography in predicting precipitation.

## 3 DATASET DESCRIPTION AND PROCESSING

The proposed RuDar dataset contains measurements from 30 weather radars mainly located in the European part of Russia (Figure 1). Radars scan the area around with a radius of 250 kilometers.

Table 1: Comparative table of various international open sources of radar data. The data were compared by spatio-temporal (km per pixel / minutes) and pixel resolution, as well as by time periods, geographical coverage and the number of radars involved. The main products contained in the datasets are base and composite reflectivity, as well as derived products: precipitation, maximum reflectivity and vertically integrated liquid (VIL). If there is more than one radar, all except NEXRAD and RuDar combine measurements from different radars into one frame.
(*) For the 2020.03–2021.02 period, we publish every fifth day to be used for validation and testing.
(**) Feature codes: (1) precipitation rate, (2) radial velocity, (3) reflectivity by altitude layers, (4) integrated or one layer reflectivity, (5) satellites

| Dataset | Time periods | Spatio-temporal resolution | Pixel resolution per frame | Geography | No. of radars | Features | Ready to use |
|---|---|---|---|---|---|---|---|
| NEXRAD Ansari et al. (2018) | 1994– | 1 km / 5 min | 460×460 | United States | 160 | 1, 2, 3, 4 | – |
| MRMS Zhang et al. (2016) | 2014– | 1 km / 2 min | 3500×7000 | United States | 160 | 1, 3, 4 | – |
| Radarnet Fairman et al. (2017) | 1970– | 1 km / 5 min | 1536×1280 | United Kingdom | 15 | 1, 3 | – |
| HKO-7 Shi et al. (2017) | 2009–2015 | 1.06 km / 6 min | 480×480 | Hong-Kong | 1 | 4 | + |
| KNMI Overeem & Imhoff (2020) | 2008–2018 | 1 km / 5 min | 400×400 | Netherlands | 2 | 4 | + |
| TAASRAD19 Franch et al. (2020) | 2010–2019 | 0.5 km / 5 min | 480×480 | Italian Alps | 1 | 4 | + |
| SEVIR Veillette et al. (2020) | 2017–2019 | 1 km / 5 min | 384×384 | United States | 160 | 4, 5 | + |
| RADOLAN Ramsauer et al. (2018) | 2014.12–2017.11 | 1 km / 5 min | 900×900 | Germany | 18 | 1, 4 | + |
| RuDar (Ours) | 2019.03–2021.02* | 2 km / 10 min | 252×252 | Russia | 30 | 1, 2, 3 | + |

Measurements come from radars every 10 minutes and have a spatial resolution of $2 \times 2$ kilometers. The dataset covers two years from 2019 to 2021 and has over 50 000 unique timestamps.

Each data sample is a three-dimensional tensor that contains the result of a 10 minute scan of the atmosphere with a single radar. The center of the frame corresponds to the location of the radar in the scanned area. Tensors have 21 channels with a spatial resolution of $252 \times 252$ pixels. The first channel contains information about precipitation intensity (mm/h) on a 600 m altitude level. The channels 2-11 contain reflectivity measurements (dBZ) performed at 10 altitude levels from 1 km to 10 km with 1 km step, and the channels 12-21 contain 10 radial velocity (m/s) measurements from the same 1-10 km altitudes. We also provide data on orography Becker et al. (2009) and latitude-longitude coordinates of the territory surrounding the area of the radar measurements. Coordinates were initially set on a kilometer grid with a step of 2 kilometers with zero at the radar location.

The data example is partially shown in Figure 2a. Please refer to the Section D in Supplementary for the comprehensive documentation[2].

### 3.1 PRECIPITATION INTENSITY

Precipitation intensity is measured at an altitude of 600 meters in mm/h and is represented by a single channel in the data. It is not a direct measurement: the precipitation rate is calculated from reflectivity values with Marshal-Palmer type of Z-R relationship Marshall & Palmer (1948). In addition to mm/h values, two special values are presented in data: `-2e6` value marks areas where measurements are not available, `-1e6` value marks areas where no precipitation events were detected. Both values can be reduced to 0 mm/h, however, when feeding a radar frame to the model, it is better to mask `-2e6` values so that the model can distinguish the absence of precipitation from the blind zone of the radar.

Figure 2d shows the distribution of precipitation by seasons. The data for each season is taken from the entire period 2019-2021. As can be seen in Figure 2d, the intensity has a seasonal dependence: the highest values of precipitation amount are reached in summer and the lowest in winter.

We investigated the distribution of precipitation intensity difference between two adjacent radar images. As Figure 2e shows, in the winter season the precipitation rate difference between two adjacent radar images is much lower than in the summer season. This means that in winter precipitation changes much less in the short-term interval than in summer, and therefore it is more difficult to solve the problem of nowcasting in warmer seasons.

Peaks which are visible in Figure 2e are the result of specificity of the combination of radar-side filtration algorithms and Marshall-Palmer type of Z-R relationship.

---

[2]Data-featuring Colab Notebook is available at `URL_hidden_due_to_blind_review`.

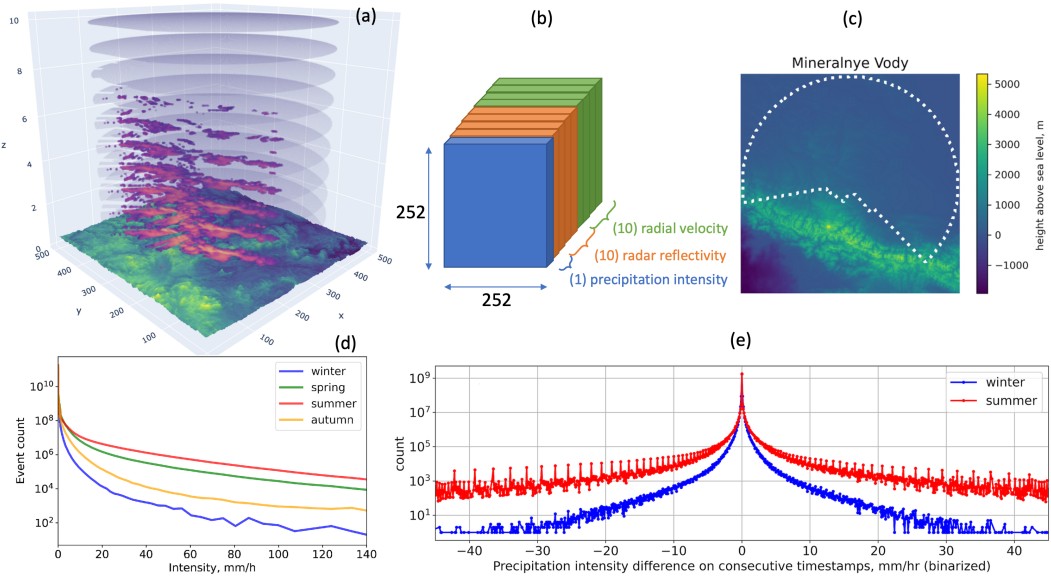

Figure 2: Featuring RuDar dataset. **(a)** The sky above the Moscow Region, Vnukovo Radar; axes ticks are given in kilometers. Ten levels of the precipitation intensity are shown (for each height level from 1 km to 10 km), together with the orography (magnified for visualization purposes) on the bottom. The lined cuts on the left part of the plot demonstrate a specific case of urban obstruction in the radar vision: skyscrapers are not shown in the orography map yet they affect the scan result. **(b)** Tensor representation for a single radar at the given timestamp: each tensor has a shape of $(252, 252, 21)$. Note that orography information is stored separately to avoid duplication. **(c)** Natural obstructions near Mineralnye Vody city: the mountain chain on the south blocks the radar, reducing the receptive field down to the area denoted by the white dotted line. **(d)** Seasonal dependence of the precipitation intensity distribution (mm/h per pixel). During the winter season events with high precipitation rates are much less represented than in the summer season. We clipped high precipitation rates ($> 140$ mm/h). **(e)** The distribution of precipitation intensity difference between two adjacent radar images (mm/h per pixel) in RuDar dataset. In the winter season, precipitation events change slower than in the summer season. The periodical peaks are the result of a peculiarity of the combination of radar-side filtration algorithms and Marshall-Palmer type of Z-R relationship. Best viewed in color.

## 3.2 RADAR REFLECTIVITY

Reflectivity values represent direct radar measurements in dBZ units. The radar measures the amount of energy reflected from droplets distributed in the atmosphere, and this amount of energy turns out to be proportional to the amount of moisture in the air. All reflectivity measurements were carried out at heights of 1-10 km with 1 km step. We provide reflectivity data at several altitude levels, since we believe that not only information about precipitation changes in time, but also in space, is important for nowcasting, see Fig. 6 in Supp. Mat. for an example.

To demonstrate the geospatial diversity, we compare the mean reflectivity for two radars: Arkhangelsk (in the north) and Krasnodar (in the south) in Figure 3. It can be seen that both seasonal and altitudinal dependencies are different for these radars, which shows the wide range of weather conditions and data variability.

## 3.3 RADIAL VELOCITY

The radial velocity is the wind velocity projected onto the ray starting in the radar. Negative values correspond to the movement towards the radar, and positive values correspond to movement away from the radar. Due to the peculiarities of this type of measurement, the values are available only at points where the radar has registered moisture droplets. The data is measured in m/s and provided

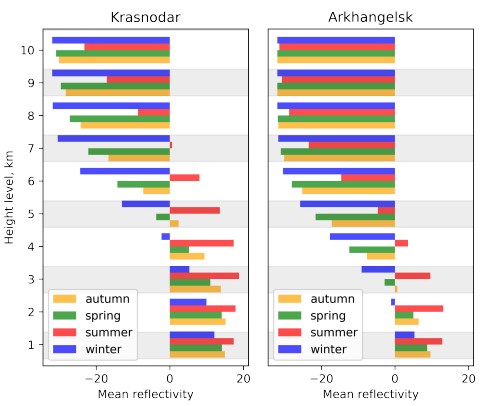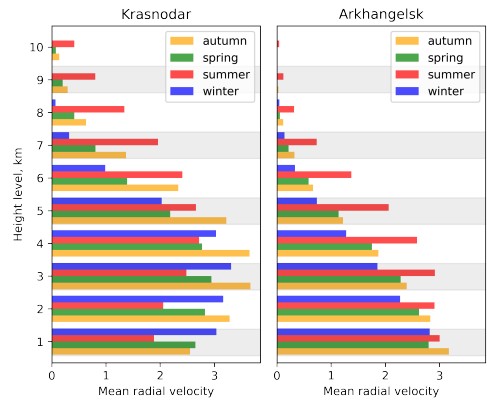

Figure 3: Comparison of two different regions: Krasnodar (on the south) and Arkhangelsk (on the north) in terms of mean reflectivity (left) and mean radial velocity (right) on different height levels. Only non-zero values are taken into account. Regions demonstrate different patterns in the altitude intensity and radial velocity distribution as well as the seasonal changes.

for ten altitude levels from 1 km to 10 km with 1 km step similar to reflectivity data. According to Figure 3, radial velocity values are seasonally dependent.

### 3.4 GEOGRAPHICAL INFORMATION

In addition to atmosphere scans, the RuDar dataset contains orography and geographical coordinates associated with each radar. Geographical coordinates (latitudes and longitudes) are provided for each point where measurements are presented. Orography is a two-dimensional tensor in which positive values correspond to elevations above sea level and negative values correspond to elevations below sea level. We resampled an available information Becker et al. (2009) to the used coordinate grid. An example of orography is shown in Figure 2c where the case of natural radar vision obstruction is demonstrated. The radar's visibility area in the figure is greatly reduced by high mountains located in the immediate vicinity of Mineralnye Vody. Note that the orography measurements do not include the urban areas, and Figure 2a shows that skyscrapers in Moscow may also pose a problem.

## 4 NOWCASTING BASELINES

### 4.1 PROBLEM STATEMENT

The nowcasting problem can be formulated in terms of sequence prediction task (precisely video prediction task) – the goal is to predict a sequence of future observations from an input sequence of historical measurements. More formally, let us have an input sequence of measurements $X = (X_{t-M+1}, \ldots, X_t)$ of length $M$ for time step $t$, and a sequence of future measurements $Y = (Y_{t+1}, \ldots, Y_{t+K})$ of length $K$. Then our task is to construct a model $f$ that predicts $Y$ by $X$:

$$Y_{t+1}, Y_{t+2}, \ldots, Y_{t+K} = f(X_{t-M+1}, \ldots, X_{t-1}, X_t). \quad (1)$$

Each element $X_i$ is a tensor of shape $H \times W \times C_{input}$, where $H$ and $W$ are the height and width of one frame, and $C_{input}$ is the number of measurements used for prediction. An element $Y_i$ of the output sequence is a tensor of shape $H \times W \times C_{output}$, where $C_{output}$ is the number of predicted measurements. The numbers $C_{input}$ and $C_{output}$ may differ, since in the input sequence one can possibly use all the available measurements (intensity, reflectivity, radial velocity, orography), and the output sequence will contain only the target values. In our case, the target is future precipitation intensity, which implies $C_{output} = 1$.

The quality of the model can be measured by how well it predicts the strength of precipitation directly, as well as how well it copes with detecting precipitation. In the first case, we use pixelwise mean squared error (MSE) to measure quality, in the second we use $F_1$-score, see details of calculation in

Section I of Supplementary Material. In meteorological articles Espeholt et al. (2021); Ravuri et al. (2021); Bouget et al. (2021), metrics such as Probability of Detection (POD), Success Ratio (SUCR), Critical Success Index (CSI) are sometimes used, but they are actually the equivalents of recall, precision and intersection over union (IOU), – standard metrics in machine learning. We chose $F_1$ because it correlates with IOU and depends on both precision and recall.

## 4.2 EXPERIMENTAL SETUP

### 4.2.1 DATA SETUP

We are focusing on a year scale setup, which uses 2019 data for training, and every fifth day of 2020–2021 data for validation and testing.

In our experiments, we use 4 input tensors with historical observations to predict 12 future intensity frames ($M = 4$, $K = 12$, $C_{output} = 1$). The spatial size of the input and output tensors is $H = W = 252$. We clip precipitation intensity values to 50 mm/h, since high intensity rates are rare in the given geographical area. As we use an MSE loss to train our models, we use a binary mask for the special value $-2e6$ to prevent penalizing our models for errors outside the radar visibility range.

When only intensity is used as an input observation, $C_{input} = 1$. Using reflectivity levels increases $C_{input}$ to 11. All values are clipped in the interval between $-32$ dBZ and 65 dBZ, and then reduced to the interval from 0 to 1 by subtracting $-32$ dBZ and dividing by $65 - (-32) = 87$ dBZ. The intensity values for this setup are divided by 50 mm/h (for both input and target frames) to be also in the unit range $(0, 1)$. Radial velocity also adds 10 input channels ($C_{input} = 11$). All radial velocity values are clipped between $-63$ m/s and 63 m/s and then divided by 63 m/s to be in the interval $(-1, 1)$. Orography adds one additional channel ($C_{input} = 2$). Negative values of heights are reduced to zero, and positive values are divided by the maximum height equal to 5336 m.

For both radial velocity and orography, the intensity preprocessing is the same as in the absence of additional features.

### 4.2.2 MODELS AND SETUPS

We represented $f(\cdot)$ with a persistent model as a weak baseline and with a state-of-the-art optical flow approach Ayzel et al. (2019) as a strong baseline. Also we trained UNet-like Veillette et al. (2020) and ConvLSTM-like Shi et al. (2015) neural network models (including Extended ConvLSTM architecture shown in Fig.4b) and tested a pretrained GAN-like model Ravuri et al. (2021). Each of the above models (excepting the pretrained GAN model) was trained on a single Tesla A100 80GB for approximately 70 hours. Details about model architectures are provided in Section E of Supplementary Materials.

Also we evaluated Earthformer Gao et al. (2022) on the proposed dataset using official implementation. Other strong neural network baselines Sønderby et al. (2020); Klocek et al. (2021); Espeholt et al. (2021) will be considered in future research as they require additional input data that is not presented in the proposed dataset. Satellite information is used in Sønderby et al. (2020), numerical weather forecast is used in Klocek et al. (2021), and Espeholt et al. (2021) requires both.

## 4.3 RESULTS

We provide the results on $F_1$ and $MSE$ for each of the above models, as well as the results of Extended ConvLSTM using each of the additional features. We evaluate the baselines on seasonal test splits of the RuDar dataset. The results are shown in Table 2.

All baselines work better than Persistent, and among baselines without additional features, Extended ConvLSTM shows the best result in all seasons except summer: there it is inferior to Optical Flow in $F_1$.

Adding reflectivity shows the best result for all metrics and seasons, except for $F_1$ in summer (where it is inferior to Optical Flow) and in autumn (where it is inferior to Extended ConvLSTM with orography). Adding radial speed improves both $MSE$ and $F_1$ and adding orography improves $F_1$ relative to Extended ConvLSTM without additional features.

Table 2: Metrics of baseline models calculated separately for each season. The first value in the cell is $MSE$, the second value is $F_1$. Metrics are average across prediction horizons. Red highlights the maximum in the entire column, blue highlights the maximum among models without additional features.

| | Spring 2020 | Summer 2020 | Autumn 2020 | Winter 2020-2021 |
|---|---|---|---|---|
| Persistent | 0.2691 / 0.5177 | 0.6281 / 0.4975 | 0.0579 / 0.5233 | 0.0260 / 0.6425 |
| Optical Flow | 0.1936 / 0.6297 | 0.4768 / 0.6435 | 0.0384 / 0.6351 | 0.0190 / 0.6884 |
| GAN | 0.1885 / 0.6368 | 0.5820 / 0.6140 | 0.0352 / 0.6507 | 0.0218 / 0.6580 |
| U-Net | 0.1420 / 0.5649 | 0.3215 / 0.5406 | 0.0312 / 0.5730 | 0.0160 / 0.6232 |
| ConvLSTM | 0.1217 / 0.6222 | 0.3244 / 0.5580 | 0.0242 / 0.6650 | 0.0156 / 0.6799 |
| Extended ConvLSTM | 0.1192 / 0.6436 | 0.3193 / 0.5877 | 0.0233 / 0.6814 | 0.0150 / 0.6947 |
| Extended ConvLSTM + reflectivity | 0.1133 / 0.6597 | 0.3080 / 0.6170 | 0.0219 / 0.6936 | 0.0140 / 0.7014 |
| Extended ConvLSTM + radial velocity | 0.1170 / 0.6516 | 0.3153 / 0.5970 | 0.0228 / 0.6885 | 0.0144 / 0.7038 |
| Extended ConvLSTM + orography | 0.1193 / 0.6526 | 0.3197 / 0.5968 | 0.0234 / 0.6876 | 0.0148 / 0.7016 |
| Earthformer | 0.0124 / NA | 0.0153 / NA | 0.0047 / NA | 0.0048 / NA |

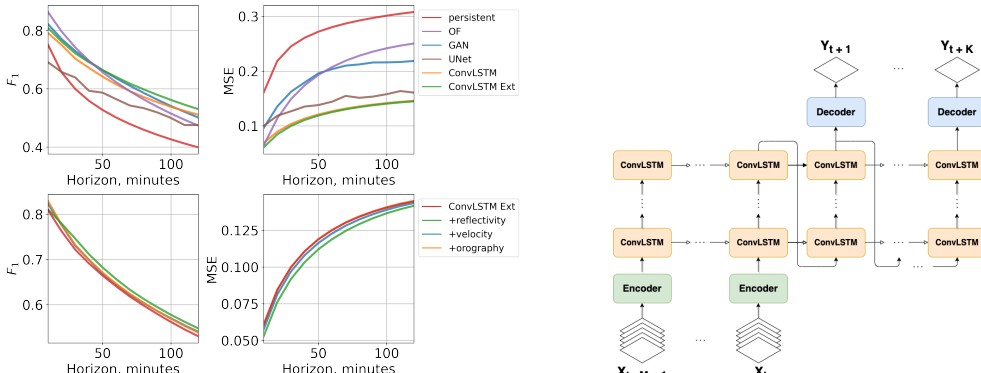

Figure 4: **(Left)** Algorithm performance (left column: $F_1$, right column: MSE) as a function of a prediction horizon on the spring season of the test set. Top row: baseline algorithms, bottom row: Extended ConvLSTM with the various additional data. While optical flow may outperform other models on the initial time steps, NN-based and trained baselines easily outperform baselines. Usage of the radar reflectivity improves the performance of the ConvLSTM model. **(Right)** An extension of the architecture from Shi et al. (2015). The first ConvLSTM layer receives the output of a convolutional encoder as input, and the output of the last ConvLSTM layer is upsampled to the final prediction with a convolutional decoder. To output the next frame, the predictor uses the outputs of the last ConvLSTM layer from the previous step.

We also show the relationship between algorithm performance and forecast horizon for the spring season in Figure 4a.

## 5 UNCERTAINTY ESTIMATION FOR THE ENSEMBLE CASE

Weather forecasting task, given its properties such as noisy data, probabilistic predictions, and general weather variability, may organically benefit from the uncertainty quantification. Uncertainty estimates (UEs) are helpful in a number of cases, including "failing to predict" (when the model understands that its prediction is not reliable enough, and gets the results from the basic robust model), active learning, data shift reaction, as well as the error analysis and model selection.

We focus on the last scenario and analyze the relationship between the error and UE for the ensemble case. Given an ensemble of five ConvLSTM models, trained on the same data but different seeds, we treat the mean and standard deviation of predictions as the output and UE, correspondingly. An ensemble error and UE on the test set are correlated with a large Spearman coefficient ($> 0.8$); this relationship holds even when both metrics are averaged over the various dimensions: time, location, and forecast horizon, see Fig. 7, 8 in Section H of Supp.Mat. We also report that we did not find a relationship between the UE and orography (Pearson $|r| < 0.1$).

This analysis suggests that UE metrics can be used to indicate large errors, which, in turn, can be used to deliver better predictions or uncertainty of the forecast, as well as assist the model selection and improvement.

# 6 DISCUSSION

## 6.1 SUMMARY

In this paper, we propose a weather radar dataset with a wide variety of geographical and climatic conditions and show in contrast to previously published works that a precipitation nowcasting task can be a seasonal dependent problem. This encourages the usage of the seasonal models and separate models for precipitation rate forecasting and precipitation events existence.

We also evaluated an ensemble of nowcasting models, analyzing the uncertainty estimates and showing its deep relation to the ensemble error on various scales.

We believe that our proposed dataset will become a reliable benchmark for precipitation nowcasting models. The code for data preprocessing and model preparation is publicly available [3]. This dataset is provided under the CC BY NC SA 4.0 license. The data is maintained on the cloud service, and is maintained by authors team[4].

## 6.2 LIMITATIONS

We would like to point out a few limitations of out work:

- **Dataset size**. We included the two-year period only in order for the dataset to be comprehensible and processible by the community, as its unpacked size already exceeds 100 GB. This dataset is not intended for a study of long-term climatic changes.
- **Data shift**. Like any other piece of real-world data, our dataset may contain several peculiarities. In the text, we discuss the seasonal dependence and geographical shifts, however, it should be noted that the measurements within the sample are also not simultaneous as they were obtained by combining the measurements from the narrow beam that scans the atmosphere.
- **Baselines**. We provide the readers with simple baselines supporting the dataset. However, we plan to benchmark a number of state-of-the-art neural network-based baselines Sønderby et al. (2020); Espeholt et al. (2021); Klocek et al. (2021), in future studies.

## 6.3 POSSIBLE PROSPECTS

A two-year dataset may be applicable not for the nowcasting task only but in a number of contemporary ML problems, including:

**Rare event detection.** Various storms and rare weather conditions are of special interest both to the researchers and end-users. Some works even emphasize this area of research Veillette et al. (2020). We expect various anomaly detection algorithms to be of great use here.

**Data shift and uncertainty estimation.** The variety of both geographical and temporal conditions makes this dataset a good candidate for the modeling of distribution shift scenarios (see, e.g. Malinin et al. (2021)), where test data may naturally vary from the training data dramatically. In this case, various uncertainty estimation approaches, like in Grönquist et al. (2019) may be helpful.

**Active learning.** One of the most important scenarios for the day-to-day forecasting systems is active learning since continuous data flow allows for the production model to learn from its own mistakes, correcting previous predictions. However, processing and retraining on huge amounts of data poses a challenge, and one may use smarter ways of data sampling (of both past archives and daily chunks of data) in order to reduce the data processing and model training times.

---

[3]The link to the full dataset will be available after paper acceptance, a sample is available at `URL_hidden_due_to_blind_review`.

[4]Please contact corresponding authors if you have any questions or additions regarding the dataset.

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

## 7 SUPPLEMENTARY MATERIALS

### A SUMMARY AND DISCUSSION OF PREVIOUS REVIEWS

We would like to thank anonymous reviewers for their valuable comments and suggestions. The main difference between the previous submission and the current one is that we shifted our focus from the ML models to the dataset, and thus provide basic and simple baselines to follow in future use. The main points raised by reviewers are listed below together with our comments:

- There is no major difference between the proposed dataset and previous datasets like HKO-7 and SEVIR. The outstanding difference – a large number of channels – needs to be justified and is straightforward to incorporate into the existing ConvLSTM-like models.
  *We emphasized the difference between our datasets and several others, including HKO-7 and SEVIR. We discuss the benefits of the additional channels in the main text.*

- The importance of this specific type of data for the ML community, not only meteorologists.
  *We added a subsection in the summary section that discusses this issue. We argue that this dataset may be of great interest to ML researchers for anomaly detection and domain shift tasks, as well as active learning and uncertainty estimation.*

- Choices on data processing and data split should be explained in detail, as well as the choice of the baselines.
  *We refined and reasoned the data processing and split scheme and updated the text. As we shifted the focus of the paper to the dataset, we now provide the most straightforward and ready-to-go baselines for comparison and future research.*

- Paper lacks critical reflection on the limitations of the proposed dataset.
  *We added discussion of the limitations to the last section of the main text.*

### B THE CHOICE OF THE BINARIZATION THRESHOLD

Initially, it was chosen upon our internal experiments within the company, which involved user's feedback for a weather forecasting service, and is close to the boundary of the rain versus moisture discrimination, given the following reasoning. Table 1 shows the number of events in the first two bins from the seasonal precipitation distribution shown in Figure 2c of the main text. One event here is one pixel in a radar frame with intensity measurements, and season data is combined for all occurrences of a given season between 2019 and 2021. For each season, the first two columns contain the number of events when there is no precipitation at all and when the precipitation strength is in the range from 0 to 0.1 mm/hr. The last column shows the proportion of no precipitation events (the sum of the first two columns) relative to all events in the season. For summer and autumn, the threshold of 0.1 mm/hr turns out to be the 95-percentile of the distribution. Since the frame-by-frame variability is much higher in summer than in winter (see Figure 2d in the main text), the summer season is more challenging for the nowcasting problem. Therefore, when choosing a threshold for binarization to obtain $F_1$ results, we focused on the summer season.

| Season | No-precipitation events | Precipitation events in range (0, 0.1] | Total events | Ratio |
|--------|------------------------|----------------------------------------|--------------|-------|
| Winter | 72,898,929,129 | 2,963,366,689 | 81,044,347,514 | 0.936 |
| Spring | 174,324,614,608 | 3,781,612,615 | 191,773,945,439 | 0.929 |
| Summer | 175,629,693,691 | 1,852,301,657 | 186,500,102,531 | 0.952 |
| Autumn | 170,587,211,782 | 4,187,861,067 | 183,448,937,655 | 0.953 |

Table 3: Seasonal comparison of the number of events with the absence of precipitation or with the strength of precipitation from the interval (0, 0.1] mm/hr, and their cumulative proportion among all events in the season.

### C RAINYMOTION LAUNCH CODE LISTING

We have added a Rainymotion Ayzel et al. (2019) launch code, which basically demonstrates that we use the default setting in our experiments.

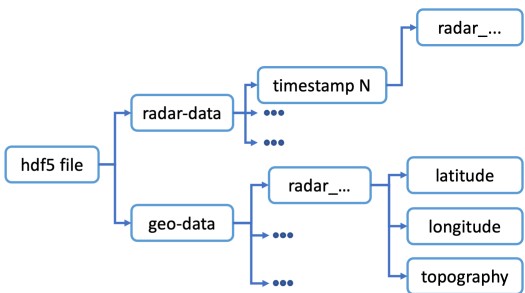

Figure 5: The scheme of an hdf5-dataset with RuDar data. Upper level – hdf5-groups with radar data and geo-data. Geo-data is arranged in the `geo-data` group by radar and contains the topography of the radar and the latitude-longitude coordinates of its territory. Radar data within the `radar-data` group is decomposed into timestamps in the form of UTC-timestamps. Data for each radar has a shape (252, 252, 21), where the first channel of the last dimension corresponds to intensity, 1-11 – to reflectivity, 12-21 – to radial velocity.

```
# workaround OpenCV 4.*
import cv2
cv2.optflow.createOptFlow_DIS = cv2.DISOpticalFlow_create
import numpy as np
from rainymotion.models import Dense

# assuming x_batch contains the necessary tensor
model = Dense()
model.input_data = x_batch    # SEQ x H x W
model.input_data[np.where(model.input_data < 0)] = 0
pred = model.run()
```

## D  DATASET STRUCTURE

The dataset is represented with a set of `hdf5` files [5]: one `hdf5` file per year. We also share subsets of RuDar dataset used in data analysis and experiments.

Each `hdf5` file consists of two groups: `radar-data` and `geo-data`. `radar-data` group provides measurements (precipitation intensity, reflectivity, and radial velocity) indexed by pairs (timestamp, radar). `geo-data` group contains the topography Becker et al. (2009) of the radar and the latitude-longitude coordinates of its territory

Figure 5 shows the schema of an `hdf5`-file.

`radar-data` group contains subgroups where each subgroup is named after a certain timestamp in UTC time-zone. Timestamps have a 10 minute temporal resolution. Each timestamp subgroup is divided into named `hdf5`-datasets with measurements from weather radars that were currently available (sometimes radars are turned off due to problems). Each radar dataset is a three-dimensional tensor that contains the result of a ten-minute scan of the atmosphere with a certain radar. The measurements include precipitation intensity (mm/hr) on the 600 m altitude (the 1st channel in a dataset), reflectivity (dBZ) (channels 2-11) and radial velocity (m/s) (channels 12-21) on 10 altitude levels from 1 km to 10 km with a 1 km step and a 2 km spatial resolution.

`geo-data` group contains subgroups where each subgroup corresponds to a certain weather radar. Each radar subgroup consists of three two-dimensional datasets with latitudes, longitudes, and topography Becker et al. (2009) of the radar territory.

---

[5]The link will be available after review

## E    MODEL INFORMATION

**Persistent.** In the persistent model, we consider the latest radar image from an input sequence as a forecast for all $K$ images of an output sequence, so it is simply a constant prediction.

**Optical Flow.** We take a state-of-the-art optical flow approach from Rainymotion library Ayzel et al. (2019), and use Dense Inverse Search model with constant-vector advection scheme. The advantage of this particular optical flow approach over the others was shown in previously published works Sønderby et al. (2020) and our experiments. See code listing for usage in Section C of Supplementary.

**GAN-like.** We used a pretrained GAN-based model from Ravuri et al. (2021) as a model for comparison, and we did not additionally fine-tune or train this model on our data. The model, as it is GAN-like, consists of a generator and discriminator parts. The generator contains two parts: encoder and decoder. An encoder is a fully-convolutional neural network that separately processes each input frame and concatenates the resulting representations. A decoder is a recurrent neural network with ConvGRU cells Siam et al. (2017) for predicting future frames. The model contains two discriminators: one for temporal consistency between predicted frames and another for spatial consistency inside a certain frame.

**UNet.** We use the same architecture as proposed in Veillette et al. (2020). It is based on the original UNet architecture Ronneberger et al. (2015) which takes $M$ input frames concatenated along the channel axis and predicts $K$ output frames also concatenated along the channel axis. The model contains four downsampling and four upsampling convolutional blocks – (32, 64, 128, 256) and (256, 128, 64, 32) filters respectively. The last layer is a convolution with one filter for producing the requested number of output channels. We trained the model for 10 epochs with $L_2$-loss and Adam optimization algorithm with a learning rate equal to $4e - 3$ and batch size $64$. For evaluation, we took a checkpoint from the best epoch according to the validation loss.

**ConvLSTM.** We used the ConvLSTM and an encoder-predictor architecture proposed in Shi et al. (2015), but with a few changes. Firstly, we do not concatenate predictions from all layers of the predictor but use only the last layer. Secondly, we use the Adam optimizer instead of RMSProp. Third, we use $L_2$-loss instead of cross-entropy. And finally, we use teacher-forcing with probability decreasing with the iteration number, similar to what authors of Wang et al. (2018) did in their implementation.

For uncertainty estimation, we trained an ensemble of five models with (128, 64, 64) filters for each layer, respectively. We trained these models with batch size 64 and learning rate $4e - 3$ for one epoch and then selected the best checkpoint for each according to the values of the validation loss.

**Extended ConvLSTM.** To analyze the impact of additional features, we expanded the original architecture with convolutional blocks at the input and output of ConvLSTM layers (see Figure 4b). We trained five models to provide an ablation study: a model without expansion by convolutional blocks, an extended model that accepts only intensity as input, and three more models, each of which accepts intensity and one of the additional features as input: reflectivity, radial velocity and orography. Each model consists of three ConvLSTM layers with (64, 64, 64) filters for each layer, respectively. We trained models for one epoch with the choice of the best checkpoint based on the values of the validation loss. We used batch size 16 and a learning rate $1e - 3$ for these experiments.

**Earthformer**. We use a transformer-like architecture called Earthformer Gao et al. (2022). The idea of the model is dividing the input tensors to cuboids and calculating self-attention inside them. After that cuboids are merged also with self-attention mechanism using additional global vectors. We have used an official implementation with hyperparameters from the original paper.

## F    FIGURES: FEATURING DATASET PECULARITIES

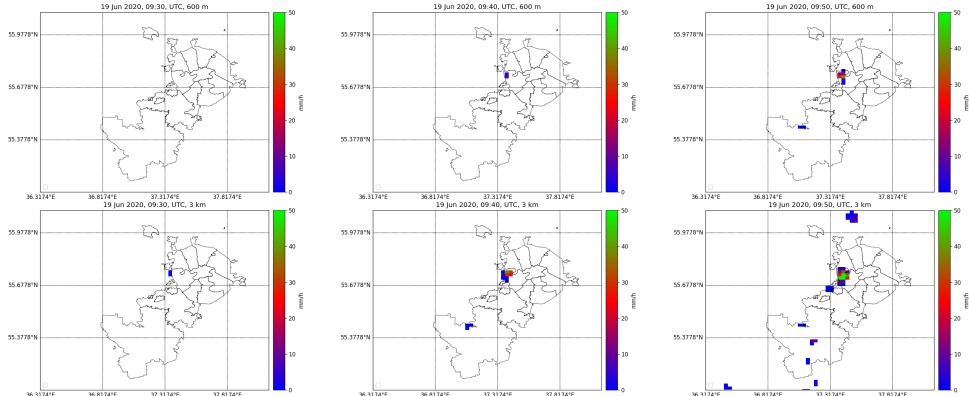

Figure 6: An example of sudden precipitation occurrence. The image shows radar measurements of precipitation rate ($mm/h$) at ground level (up to 600 meters, top row) and reflectivity ($dBZ$ converted to $mm/h$ by Marshall–Palmer relation) at an elevation of 3 km (bottom row) for three consequent time moments with an interval of ten minutes. The data from a 3 km height provide information about future precipitation before the actual rain starts. The example is for June 19th, 2020, 9:30 AM UTC, Moscow, Russia. The color in the pictures corresponds to the levels of precipitation intensity, which may vary from 0 (blue) to 50 mm/h (light green).

## G  UNCERTAINTY ESTIMATION DETAILS

Here we provide an additional details to the uncertainty estimation (UE) section in the main text.

For a given pixel value at the position $(x, y)$ for radar $r$, time horizon $k$, and time $t$ we can define a ground truth $y_{(r,k,t,x,y)}$. Let us have five predictor models $f_1, \ldots, f_5$ with the corresponding predictions $\hat{y}_{(1)}, \ldots, \hat{y}_{(5)}$ (we omit indices here for simplicity). We define an ensemble prediction as $\hat{y}_{(r,k,t,x,y)} = \frac{1}{5} \sum_i \hat{y}_{(i)}$, and the (biased) standard deviation (which we treat as an UE here) as $UE_{(r,k,t,x,y)} = \frac{1}{5}(\sum \hat{y}_{(i)}^2 - (\sum \hat{y}_{(i)})^2)$. We note that while we do not use an unbiased estimate, it is the same up to the scaling factor, and does not affect the calculated correlations. An ensemble error is defined as $AE_{(r,k,t,x,y)} = |y_{(r,k,t,x,y)} - \hat{y}_{(r,k,t,x,y)}|$.

We use the straightforward SciPy implementations for the Pearson and Spearman correlations, which, in turn, correspond to two different usage scenarios. In the first scenario, we can actually approximate an error with UE if we believe that the linear relationship (indicated by Pearson correlation) is strong enough for this region, season, or radar. The second scenario, which uses a rank Spearman correlation, is more suitable for the outlier detection and other sample ranking activities, like acquisition function for an active learning.

While aggregation performed for the Fig. 8 is pretty straightforward, aggregation for the Fig. 7 was performed as follows. First, we perform the averaging of ensemble error and UE grouped by horizon, radar, month, and hour over the remaining dimensions (see pseudo-SQL code listing below). After that, we calculate the correlation between the vectors of UE and ensemble error corresponding to the graph points for both left and right plots. All the correlation values we provide are statistically significant.

```
SELECT
    AVG(ensemble_error) AS avg_ae,
    AVG(stdev) AS avg_ue,
    horizon, radar, month, hour
FROM (
    SELECT ensemble_mae, stdev, horizon, radar, month, hour
    FROM pixelwise_error_table
)
GROUP BY
    horizon, radar, month, hour
```

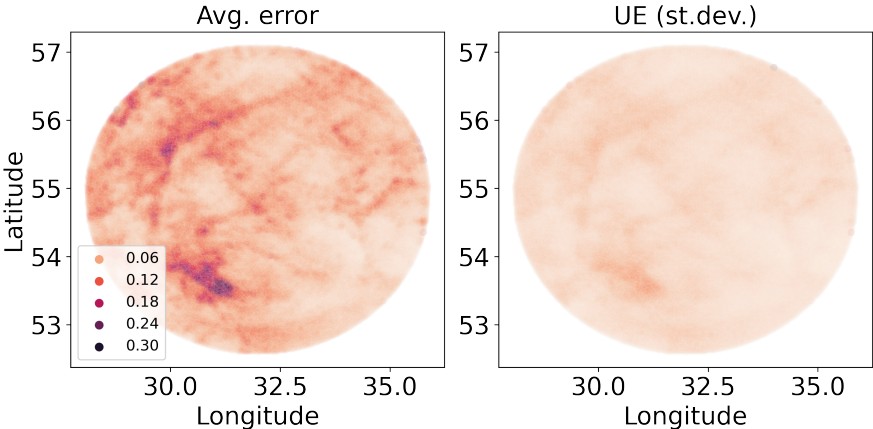

Figure 7: Error and UE for the Smolensk radar averaged pixelwise over the summer season for the 10 min horizon. Large error regions are located at the same coordinates as large UE values.

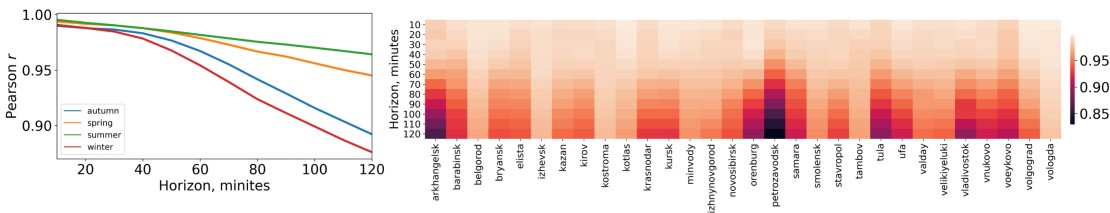

Figure 8: UE for the ensemble case. **Left:** Pearson correlation coefficient between the error and UE, averaged along the radars and time, as a function of the forecast horizon. Autumn and winter seasons are less predictable with the uncertainty quantification. **Right:** Heatmap of the correlation coefficient as a function of radars and horizon. Northern radars' errors correlation with UE is smaller.

## H FIGURES: UNCERTAINTY ESTIMATION

## I    METRICS CALCULATION DETAILS

We use raw mm/h values to calculate $MSE = \sum_{i=1}^{H \times W} (y_i - \hat{y_i})^2 / (H \times W)$ for a single frame, where $y_i$ is a pixel value from the ground truth sequence and $\hat{y_i}$ is the corresponding value from the predicted sequence. To calculate $MSE$ on the entire dataset, we iterate over the dataset and maintain a vector of length $C_{output}$ accumulating the sum of $MSE$ values for each prediction horizon. To get the final result, we first divide the values obtained in the vector by the total number of sequences and then average them.

In order to calculate $F_1$-measure, we beforehand binarize ground truth and predicted sequences with some threshold. In our experiments, we took one of the standard (see Sønderby et al. (2020)) binarization thresholds equal to $0.1$ mm/h. The metric is defined as $F_1 = (2 \cdot TP)/(2 \cdot TP + FP + FN)$, where $TP$ – cases when precipitation was present both in the predicted sequences and in the ground truth, $FP$ – cases where precipitation was present in the predicted sequences but was not present in the ground truth, $FN$ – cases where precipitation was not predicted but was present in the ground truth. To calculate $F_1$, we iterate over the dataset and maintain a vector of length $C_{output}$ accumulating $TP$, $FP$ and $TN$ for each prediction horizon. After we have gone through the entire dataset, we count $F_1$ for each position in the vector, and then average the values obtained.

