# OpenReview forum: "RuDar: Weather Radar Dataset for Precipitation Nowcasting with Geographical and Seasonal Variability"
_ICLR.cc/2023/Conference — Submitted to ICLR 2023_

### Official Review · Reviewer_ae7S · 2022-10-24

**Confidence:** 4
**Correctness:** 4
**Technical Novelty And Significance:** 2
**Empirical Novelty And Significance:** 3
**Recommendation:** 5

**Clarity, Quality, Novelty And Reproducibility:**

The paper is very well written and clear.
While a very cutting-edge database release, revealing new insights and new model usages, can be worth publishing, it looks to me that this database is too similar to the others to be publishable in ICLR. The release of such dataset is important, and it is good for the community to have access to it, but it can't be the main contribution of an ICLR paper.
Both codes and data are released, even if it is not possible to access it now (the url is hidden due to blind review)

**Strength And Weaknesses:**

Strengths:
- The dataset is rich, containing information at several altitude levels, during 2 years, and with additional fields that are not always present in such datasets.
- the comparison with other datasets is detailed, clear and very useful
- a large number of state-of-the-art deep learning methods are compared on this dataset
- a study on uncertainty estimation, performed by varying the seed of the learning phase, is performed

Weaknesses:
- from the table 1, it is not clear how this new dataset would make a difference, in particular, it looks to me that the fact that this dataset is coarse (the one with the smaller grid size yet on a large region) is probably very limiting for a task such as precipitation.
- there are no methodological novelties
- the uncertainty estimation is rather simple


**Summary Of The Paper:**

The paper presents a database of precipitation forecasting from 30 Russian meteorological stations, and a benchmark of state-of-the-art deep learning methods for the forecast estimation.

**Summary Of The Review:**

The paper is very clear and presents an interesting database and benchmark of different precipitation forecasting databases and deep learning forecast estimation models. But I don't think it is enough for an ICLR publication as there is no particular novelty on either the database nor the methods.

---

> ### Author Response · Authors · 2022-11-18
> **A reply to the review by Reviewer ae7S**
>
> Thank you for the comments.
>
> > from the table 1, it is not clear how this new dataset would make a difference, in particular, it looks to me that the fact that this dataset is coarse (the one with the smaller grid size yet on a large region) is probably very limiting for a task such as precipitation.
>
> Yes, considered weather radars have limitations in terms of both space and time discretization: they measure the atmosphere state with 2 km space resolution and 10 min time resolution – this is the result of hardware limitations we cannot overcome. However, in contrast to other publicly available datasets, RuDar, as a ready-to-use ML dataset, contains the combination of precipitation rate, reflectivity, and radial velocity measurements, which are not provided by other ready-to-use datasets.
>
> > there are no methodological novelties
>
> This paper is focused on describing the RuDar dataset and evaluating baseline models on it. We welcome the usage of the proposed dataset for developing better precipitation nowcasting models, and we aim at the large-venue visibility for both reasons of dataset visibility as well as the general popularization of weather forecasting and precipitation nowcasting as a challenging ML task.
>
> > the uncertainty estimation is rather simple
>
> We do not provide a detailed error analysis for the uncertainty estimation section as it depends on a number of factors, including the model choice (we used the ConvLSTM as the baseline here, not adding any additional features), data, model training options, etc. While this paper is mainly focusing on featuring the RuDAR datasets and setting up simple yet contemporary baselines, we do believe that deeper analysis of the UE-error relation is an interesting topic for the future studies.

---

### Official Review · Reviewer_MvGQ · 2022-10-25

**Confidence:** 3
**Correctness:** 3
**Technical Novelty And Significance:** 3
**Empirical Novelty And Significance:** 2
**Recommendation:** 6

**Clarity, Quality, Novelty And Reproducibility:**

Could you provide an evaluation of the quality, clarity and originality of the work?
The proposed dataset is significantly important, which will be practical in precipitation nowcasting.
The method could be easily implemented follow their descriptions.


**Strength And Weaknesses:**

Strength:
Their proposed weather radar dataset considers more dimensions in contrast to previously published works, which could be useful to achieve the precipitation nowcasting. This dataset covers multiple climate zones, large timestamps over a two-year period.

Weaknesses:
The baseline methods seem too naïve and lacks more ablation study experiments.
1.	The authors claim their dataset could be applied into other ML task, e.g. data studying, anomaly detection. Wish to conduct related experiments to verify this statement.
2.	I wonder whether there exist additional data from other zones, which could be implement the cross evaluation of their baseline models.
3.	There seems lack more details related works about this topic.
4.	The evaluation metric just uses MSE in Table 2, which cannot reflect the accuracy of nowcasting.
5.	The results of the previous methods on the proposed dataset also should be listed.


**Summary Of The Paper:**

This work builds a new weather radar dataset to explore the precipitation nowcasting task, which additionally considers a wide variety of geographical and climatic conditions.  Moreover, the authors carefully describe the acquisition process and distributions of their dataset. Then, they perform several baselines for evaluation on their proposed dataset.

**Summary Of The Review:**

I suggest the authors could submit to a more appropriate journal will have a higher impact in their community.

---

> ### Author Response · Authors · 2022-11-18
> **A reply to the review by Reviewer MvGQ**
>
> Thank you for your questions and comments.
>
> > The authors claim their dataset could be applied into other ML task, e.g. data studying, anomaly detection. Wish to conduct related experiments to verify this statement.
>
> We think it is a good and interesting area for the following research, but probably it is out of the paper scope as is, especially regarding the page limit constraints.
>
> > I wonder whether there exist additional data from other zones, which could be implement the cross evaluation of their baseline models.
>
> We think that experiments with domain adaptation and evaluation on out of domain data (e.g. train the model on RuDar, and evaluate it on MRMS, or train the model on RADOLAN, and fine-tune it on TAASRAD19) could be interesting and valuable research directions in the near future. However, we would like to note that historically the data produced by the weather radars in various countries is different, starting from the measuring methodology and hardware calibration techniques, and ending with a variety of byproducts that the processing software outputs. Therefore, different data is not directly comparable and such a task deserves another deep and thorough study.
>
> > There seems lack more details related works about this topic.
>
> The paper is mostly about the dataset, and we considered most of the up-to-date works that propose precipitation nowcasting datasets, so we claim our literature review (Section 2 in the main text) to be round and inclusive. Moreover, in terms of nowcasting models, we described (Section E in the Appendix) and evaluated common baselines in the precipitation nowcasting area, and also added experimental results of Earthformer [1]. Additional Earthformer ablation study on our dataset will be included in the camera-ready version of the paper.
>
> > The evaluation metric just uses MSE in Table 2, which cannot reflect the accuracy of nowcasting.
>
> Table 2 also contains evaluation results in terms of F1 measure. In our opinion, MSE provides information about correctness of intensity rate forecasting, and F1 shows the model quality in terms of precipitation presence. We have chosen F1 because it is widely used in the machine learning community and it strongly correlates with IoU (CSI) — another measure commonly used in the precipitation nowcasting literature. The chosen threshold for calculating F1 is 0.1 mm/hr, as a one of common thresholds in precipitation nowcasting evaluation.
>
> > The results of the previous methods on the proposed dataset also should be listed.
>
> We included experimental results of baseline approaches (described in the Section E in the Appendix), which are widely used in the precipitation nowcasting area. Additionally we evaluated Earthformer [1] on our dataset with intensity only inputs. Earthformer experimental results with added reflectivity and radial velocity information will be included in the camera-ready version of the paper.
>
> [1] Earthformer: Exploring Space-Time Transformers for Earth System Forecasting [Gao et al.; NeurIPS 2022]

---

### Official Review · Reviewer_TLZx · 2022-10-25

**Confidence:** 4
**Clarity, Quality, Novelty And Reproducibility:** 1) The paper is clear and easy to und…
**Correctness:** 4
**Technical Novelty And Significance:** 2
**Empirical Novelty And Significance:** 2
**Recommendation:** 5

**Strength And Weaknesses:**

Pro:
1) The dataset is attractive, I wonder how the models performance across different datasets, are their performance rankings consistent or not ?

Cons:
2) The methods used are relatively low-cost, I wonder how heavy cost models, such as transformers like architectures, perform comparing with these baselines.

How is the dataset usability ?  Is the model trained based on such dataset can be comparable with the most advanced weather forcast devices .



**Summary Of The Paper:**

1) This paper propsed a different weather prediction model by inducing a new dataset [above russia] which is adopted to evaluate all current models such as ConvLSTM etc.

2) The results looks promising on this new datasets.

**Summary Of The Review:**

Good dataset, while limited domain relativity and novelty.

---

> ### Author Response · Authors · 2022-11-18
> **A reply to the review by Reviewer TLZx**
>
> Thank you for the comments.
>
> This paper presents the weather radar dataset, and the main intended users are weather forecasting practitioners, who employ the machine learning techniques as a part of their work, as well as deep learning enthusiasts. Therefore, we evaluated classic deep learning baselines, which are used in the precipitation nowcasting area, and, according to the similar studies [1], are suitable for the comparison and are not trivial to beat.
>
> > The methods used are relatively low-cost, I wonder how heavy cost models, such as transformers like architectures, perform comparing with these baselines.
>
> Thanks for your comment. We have made an additional evaluation of (a very recent) Earthformer [2] model on the proposed dataset. Due to time constraints, we added experimental results of the setup only with intensity inputs. Wider Earthformer ablation study will be included to the camera-ready version of the paper.
>
> [1] SEVIR: A Storm Event Imagery Dataset for Deep Learning Applications in Radar and Satellite Meteorology [Veillette et al.; NeurIPS 2020]
>
> [2] Earthformer: Exploring Space-Time Transformers for Earth System Forecasting [Gao et al.; NeurIPS 2022]

---

### Author Response · Authors · 2022-11-18
**Thank you for your time and comments**

We would like to thank all reviewers for their valuable comments. To extend the evaluation with the newest and more heavy-cost models, we have updated the text with Earthformer [1] experiment results. Unfortunately, we could evaluate the setup only with intensity input (mm/hr) to the rebuttal deadline. The results with added reflectivity, radial velocity, and orography information will be available in the camera-ready version.

[1] Earthformer: Exploring Space-Time Transformers for Earth System Forecasting [Gao et al.; NeurIPS 2022]

---

### Decision · Program_Chairs · 2023-01-20

**Decision:**

Reject

**Justification For Why Not Higher Score:**

This paper cannot satisfy the criteria for reviewing papers that focus on datasets and benchmarks.


**Justification For Why Not Lower Score:**

N/A


**Metareview: Summary, Strengths And Weaknesses:**

This paper is relevant to both the machine learning and meteorological communities with research interests in weather forecasting in general and precipitation nowcasting in particular. Unlike most of the ICLR submissions, the work proposes a new benchmark dataset on which multiple existing methods are evaluated and compared.

Admittedly, the general ICLR review criteria may not be the most suitable for this type of papers. In fact this is why NeurIPS has introduced the Datasets and Benchmarks Track (https://neurips.cc/Conferences/2022/CallForDatasetsBenchmarks). Their review criteria were specially designed for this type of papers and hence can be used here for reference as well.

Let me point to some factors that are considered when evaluating submissions for the NeurIPS special track: (1) Utility and quality of the submission: Impact, originality, novelty, relevance to the NeurIPS community will all be considered. (2) Completeness of the relevant documentation: For datasets, sufficient detail must be provided on how the data was collected and organized, what kind of information it contains, how it should be used ethically and responsibly, as well as how it will be made available and maintained. For benchmarks, best practices on reproducibility should be followed. (3) Accessibility and accountability: For datasets, there should be a convincing hosting, licensing, and maintenance plan.

I am afraid this paper could not be accepted if the same review criteria would be applied, particularly on the second and third points. The authors are recommended to consider them in revising their paper if they want it to be published in a suitable venue that can make high scientific impact.